# Exploring adolescent engagement in sexual and reproductive health research in Kenya, Rwanda, Tanzania, and Uganda: A scoping review

**Hanna Chidwick**[1]*, **Andrea Baumann**[1], **Patricia Ogba**[1], **Laura Banfield**[2], **Deborah D. DiLiberto**[1]

1 Faculty of Health Sciences, Global Health Office, McMaster University, Hamilton, Ontario, Canada,
2 Health Sciences Library, McMaster University, Hamilton, Ontario, Canada

* chidwihw@mcmaster.ca

## Abstract

Adolescent sexual and reproductive health (ASRH) in East Africa has prioritized research on the barriers to care, communication, and ASRH knowledge, attitudes, and practices. However, there is little research examining the extent to which meaningful adolescent engagement in research is achieved in practice and how this influences the evidence available to inform ASRH services. This review offers a critical step towards understanding current approaches to adolescent engagement in ASRH research and identifying opportunities to build a strengthened evidence base with adolescent voices at the centre. This scoping review is based on Arksey and O'Malley's (2005) framework, employing a keyword search of four databases via OVID: Medline, Global Health, Embase and PsycINFO. Two reviewers screened title, abstract and full text to select articles examining ASRH in Tanzania, Rwanda, Kenya, and Uganda, published between 2000 and 2020. After articles were selected, data was extracted, synthesized, and thematically organized to highlight emerging themes and potential opportunities for further research. The search yielded 1201 results, 34 of which were included in the final review. Results highlight the methods used to gather adolescent perspectives of ASRH (qualitative), the content of those perspectives (knowledge, sources of information, gaps in information and adolescent friendly services), and the overall narratives that frame discussions of ASRH (risky sexual behaviour, stigma, and gender norms). Findings indicate the extent of adolescent engagement in ASRH research is limited, resulting in a lack of comprehensive evidence, consistent challenges with stigma, little information on holistic concepts and a narrow framing of ASRH. In conclusion, there is opportunity for more meaningful engagement of adolescents in ASRH research. This engagement can be achieved by involving adolescents more comprehensively throughout the research cycle and by expanding the range of ASRH topics explored, as identified by adolescents.

**Data Availability Statement:** All relevant data are within the paper and its Supporting information files.

**Funding:** The authors received no specific funding for this work.

**Competing interests:** The authors have declared that no competing interests exist.

## Introduction

Adolescents make up over half of the world's population and represent the largest generation in history [1,2]. Defined by the United Nations (UN), as young people aged 10 to 19 years old, adolescence is a key time influencing the trajectory of individual and community health outcomes [1,2]. Of particular interest are adolescent sexual and reproductive health (ASRH) outcomes, as adolescents start engaging in sexual behaviour around the age of 15 years [1,3]. Sexual and reproductive health (SRH) is defined as "a wide range of health issues including family planning; maternal and newborn health care; prevention, diagnosis and treatment of sexually transmitted infections (STIs) . . . [with] services [aiming to prevent] poor SRH, such as . . . unintended pregnancies, unsafe abortions, [and] complications caused by STIs" [4,5]. Improving SRH outcomes aligns with Sustainable Development Goals (SDG) 3.7 and 5.6, which aim to ensure universal access to SRH services and rights [4,5].

In Sub-Saharan Africa, which includes East African countries such as Kenya, Rwanda, Tanzania, Uganda, progress on achieving SDG targets remains slow, resulting in a continuing burden of STIs, unmet contraceptive needs, and inadequate quality of SRH care among adolescents [4,6–13]. As such, the focus of health-specific literature in East Africa has been on SRH interventions, barriers to care amongst adolescents, parent and community communication on SRH, and knowledge, attitudes and practices [6,9,14–28]. This slow progress has been attributed to the complexity of defining ASRH priorities and inadequate research evidence to support relevant policy and program decisions [4,12,29]. Increasing adolescent involvement in research about ASRH has been suggested as a way to build a more responsive and relevant evidence base [2,29–32]. Adolescent participation can contribute to defining ASRH services, polices, and programs that better reflect adolescents needs [2,29–32].

There is little research examining the extent to which meaningful adolescent engagement in ASRH research is achieved in practice and how this influences the evidence generated to inform ASRH programming, education, policies and services [2,29–31]. This review offers a critical step towards understanding current approaches to adolescent engagement in SRH research and identifying opportunities to build a strengthened evidence base with adolescent voices at the centre.

The purpose of this paper is to examine adolescent engagement in ASRH research in four East African countries, Kenya, Rwanda, Tanzania, and Uganda. These four countries were selected based on the Organization for Economic Cooperation and Development (OECD) Development Assistance Committee data on Aid to Health, which highlights that the largest investments in reproductive health in the East African Community (EAC) are flowing to Tanzania, Uganda, South Sudan, Kenya and Rwanda [33]. However, South Sudan was not included in this analysis given its unstable context and heavy investment in the humanitarian rather than development sector over the last 15 years. In this paper, we will examine, (i) the methods used to gather adolescent perspectives of SRH, (ii) the content of those perspectives and, (iii) the narratives framing discussions of ASRH. The review was guided by the following question: How are adolescents engaged in research on ASRH and what are the perspectives about ASRH in Kenya, Rwanda, Tanzania, and Uganda?

## Methods

This scoping review aimed to map the existing literature on ASRH in Kenya, Rwanda, Tanzania, and Uganda, to consider how adolescents have been engaged in research and the related perspectives about ASRH. A scoping review allows for iterative "mapping" of existing literature towards identifying gaps for future research [34–36]. Our approach was informed by Arksey and O'Malley's (33) scoping review framework and the PRISMA-ScR Checklist by Tricco et al.

(S1 Checklist) (35). We followed a five-stage process involving, 1) identifying the research question, 2) identifying relevant studies, 3) selecting studies, 4) charting the data, and 5) collating, summarizing and reporting results [34].

### Identifying relevant studies

We searched four databases through OVID: Medline, Global Health, Embase and PsycINFO. The search strategy involved the keywords "sexual health" or "reproductive health" or "sexual and reproductive health" AND Adolescent* or teen* or youth AND "East Africa" or Tanzania or Rwanda or Uganda or Kenya (S1 Text). The review included peer-reviewed articles written between 2000 and 2020. All searches are current to February 2021.

The inclusion criteria were studies that, 1) focused on adolescents (aged 10–18, pre-university); 2) were conducted in Tanzania, Rwanda, Uganda, or Kenya; 3) discussed the concept of SRH and specifically adolescent perspectives on, from and of SRH; and, 4) were written in English and published between 2000–2020 to capture the current literature on SRH along with historical perspectives.

The exclusion criteria were, 1) literature on interventions or prevalence of SRH that did not provide adolescent perspectives on, from and of SRH; 2) grey literature; 3) studies that were implemented in countries in East Africa outside of Tanzania, Rwanda, Uganda, and Kenya; and, 4) literature that focused on "last mile populations" or niche communities (i.e., refugees). We did not exclude studies based on study design.

### Selecting studies

Two reviewers completed the title, abstract screening, and full text review through Covidence, an online reviewing platform. If there was disagreement on inclusion, reviewers discussed until consensus. The primary author subsequently completed an additional full text review for comprehension.

### Charting the data

Information from selected studies was collated and summarized using the data extraction framework (Table 1).

## Results

The search yielded 1201 results. Out of those results, 398 were excluded as duplicates and 803 articles were screened (title and abstract). Subsequently, 121 texts were reviewed fully and 34 were included in the final review (see Fig 1, S1 Data). Of these studies, 11 were conducted in Uganda [6,14,26,37–44], 12 in Tanzania [22,27,45–54], eight in Kenya [8,55–61], and three in Rwanda [10,11,62]. The results were summarized to highlight the methods used to gather adolescent perspectives of SRH, the content of those perspectives, and the narratives that frame discussions of ASRH.

### Methods used to gather perspectives

Almost 62% of articles implemented a qualitative study design and used in-depth interviews and focus groups to collect data. Quantitative study designs accounted for 26%, mixed methods for 6% and participatory approaches for almost 6%. Adolescents were included primarily as study participants and were the only participants in 25 articles [6,8,10,11,22,26,27,38,39,42–44,46–49,51,53,55,56,58,60]. In eight articles, adolescents were involved as participants alongside community members, teachers, caregivers, religious leaders, and healthcare providers

**Table 1. Data extraction framework.**

| Category | Description |
|---|---|
| Author | Where is the main author(s) from? Is the lead author from an African or non-African based institution? |
| Title | What is the title? |
| Journal & type of publication | Indicate both the journal and type of publication (i.e. peer reviewed article etc.) |
| Year of publication | What is the year of publication? |
| Aim/objectives of paper | Describe aim/objective of study |
| Overview of study | Country of study focus, type of study, intervention examining (if applicable), thematic focus area (identify the problem being studied), main objective/aim of study |
| Study design and methods | Type of study, methodology, approach, data collection tools, analysis, outcome measures<br>Whose perspectives were included, discussed, and gathered in study? What do those perspectives involve?<br>Were adolescents involved in development of research, collection of data/analysis, used as member checkers? |
| Overview of results/ conclusions | Reported outcomes, summary of key findings, highlight conclusion |
| Limitations | Describe study limitations |

[14,40,45,50,52,54,57,59,62]. One study captured adolescent perspectives without directly involving adolescents [41]. None of the articles indicated that adolescents had contributed to developing the research design or contributed to the analysis of data (i.e., member checkers, verifying data analysis, etc.).

## Content of perspectives

The content of adolescent perspectives of SRH can be specified as knowledge, sources of information, gaps in information, and adolescent friendly services.

**Knowledge.** Thirteen articles examined ASRH knowledge, including concepts such as family planning, contraceptives (i.e., condoms), STIs, puberty and, abstinence [8,10,11,22,26,37,39,40,42–45,50,51]. Most of these articles quantitatively measured level of SRH knowledge among adolescents and how knowledge translates into behaviour. Based on the findings, a high level of SRH knowledge did not translate into safer sexual behaviour or genuine understanding of SRH [40,44,45,55,58].

**Sources of information.** Six articles identified the primary sources of SRH information among adolescents as peers, radio, and parents, with some mentioning health centres [8,14,22,39,47,50]. Radio was identified as the most common source of information due to mass media campaigns, accessibility and convenience [22,47]. Adolescents indicated that parents were a valued and desired source of information even though many parents assumed ASRH was immoral and were hesitant to discuss the topic [14,41,43,45,50,54,59]. Due to this hesitation from parents, adolescents indicated that peers provide a significant amount of SRH information [8,14]. Although health centres were cited as a source of information, the limited confidentiality, lack of adolescent-specific information, and low trust in the care provided, deterred adolescents from seeking SRH information at health centres [22,52,59].

**Gaps in information.** Adolescents reported misconceptions about aspects of SRH such as contraceptives [38,39,42,43,49,56,60,62]. For example, a number of articles indicated that adolescents believed contraceptives, abortion and family planning were linked to infertility and cancer due to information shared by parents and communities, and due to fear regarding side effects from biomedical treatment [8,22,38,42,43,56,58,60]. The fear of infertility is connected

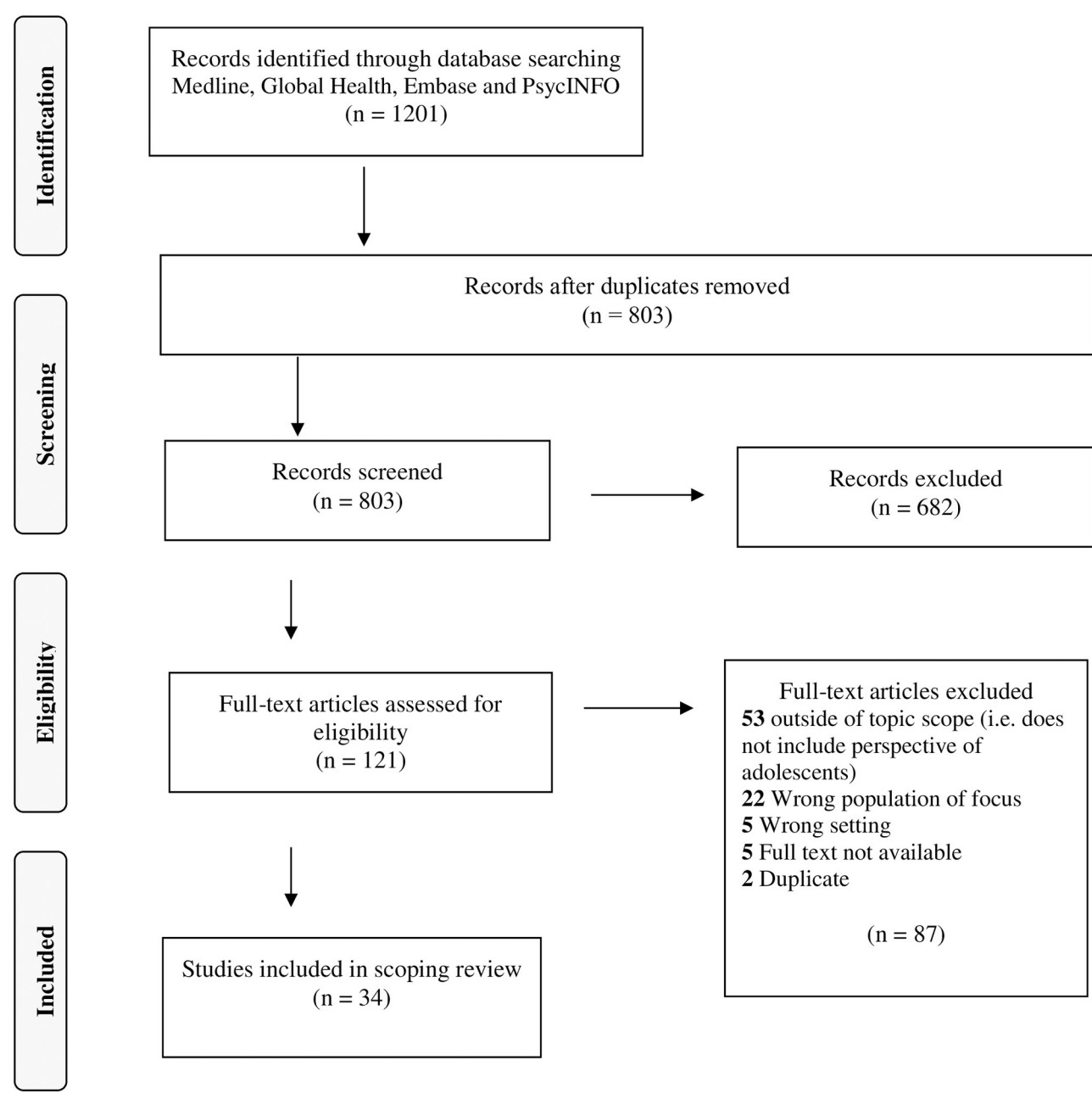

**Fig 1. PRISMA flow diagram.**

to the high value placed on having children in East Africa [43,53]. Literature noted that adolescents are constrained by cultural and gender norms which equate having children with status and value, creating pressure to prove their fertility, especially among young women [43]. Two articles discussed adolescents wanting to know about diversity in sexual behavior (i.e., curiosity about sex rather than just sexual health) [10,49].

**Adolescent friendly services.** Seven articles discussed adolescent friendly services [6,8,40,41,50,52,59]. These were defined as non-judgmental, private spaces in which care is free and health practitioners are trained on specific ASRH needs [6,8,50]. Tanzania, Rwanda,

Kenya, and Uganda have all focused on providing adolescent friendly services at existing health clinics to increase the access adolescents have to SRH information and care. However, access is limited due to barriers such as confidentiality and limited knowledge of adolescent-specific SRH (6,8,52,55,56,59). Three articles noted the need to shift towards more adolescent involved development of services to address misconceptions and mistrust in existing services [6,8,50].

### Narratives framing discussions of ASRH

Narratives framing discussions of ASRH in the literature were consistent over the 20-year period of the review and included risky behavior, stigma, and gender norms. Twelve articles discussed risky sexual behaviour, that is, adolescents engaging in sexual relations, such as sex without a condom and having multiple sexual partners, leading to negative outcomes (e.g., STIs, unwanted pregnancy) [8,11,43–45,50,52–55,58,61]. Transactional sex and a mistrust in condoms were also noted as risky sexual behaviour [43,44,52–54,61]. Sex for pleasure was considered deviant or risky behaviour among adolescents in Rwanda, where abstinence was framed as "good" sexual behaviour [10].

Some articles noted that high levels of SRH knowledge do not translate into decreased risky sexual behaviour [44,45,55,58]. Other articles highlighted how social and environmental factors such as experiences of poverty, unequal gender norms, substance abuse, and lack of parental support correlate with increased risky sexual behaviour [6,8,10,11,50,54].

Stigma, a social determinant of health specifying what is acceptable or deviant SRH behaviour based on assumptions and stereotypes (i.e., religious beliefs and assumptions of sexual "promiscuity"), was discussed in 11 articles, particularly from healthcare providers, communities and parents [38,40,42,43,46,52,53,56–58,61]. Stigma towards sex outside of marriage, use of contraceptives (especially condoms), early or unintended pregnancy, and abortion were discussed as leading to shaming, social isolation and negative name calling [8,38,42,43,52–54,56,61]. Stigma towards contraceptive use was based on the assumption that contraceptives are only for people who are married or have children, and if used otherwise, result in infertility or increased risky sexual behaviour [8,38,42,43,52,56].

Nine articles discussed how gender norms limit the ability of young women to access SRH care and/or engage in safe sexual behavior (e.g., condom use) [8,10,38,42–44,50,53,55]. This included reference to patriarchal gender norms that prioritize men over women resulting in power imbalances and decreased decision making capacity of women, while reinforcing assumptions that young women are sexually promiscuous [10,38,50].

### Discussion

With such a significant population of adolescents in the world and adolescence being a key time of growth and challenge, especially in regards to SRH, it is important to identify how adolescents have been engaged in SRH research in Kenya, Rwanda, Tanzania, and Uganda. Findings indicate the extent of adolescent engagement is limited, resulting in a lack of comprehensive evidence, consistent challenges with stigma, little information on holistic concepts, and a narrow framing of ASRH.

In the studies reviewed, adolescents were involved as study participants but not in the development or analysis of research projects. Limited engagement through top down, generalized research and programming fails to recognize the intersectional, diverse and context-specific SRH experiences of adolescents in East African countries [2,29,30,63–65]. The value of involving adolescents in SRH research, education and programming development is established [2,29,31,32,50,66–69]. The WHO has called for participatory engagement of

adolescents, supporting programs and policies that are "partnership-driven, evidence-informed, gender-responsive, human rights-based, sustainable, people-centered, [and] community-owned" [32]. Participatory action research and community-based participatory research methods have been shown to effectively engage research participants more comprehensively throughout the process of research, resulting in more equitable, context-specific research information and action [30,70–72]. In particular, methods such as photovoice, meaningfully engage adolescents leading to greater peer support, knowledge and insight to inform future SRH education and programming [30,73–75]. It is essential to further engage East African adolescents in SRH research, education and program development in order to create more effective ASRH programs [29,30].

Stigma related to ASRH was noted as a significant challenge for adolescents and stakeholders in the studies reviewed. Stigma from healthcare providers, community leaders and parents towards contraceptive use ultimately limits the perceived and experienced access adolescents have to adolescent friendly services and SRH information [38,43,52,56,61]. Research on how to reduce stigma towards ASRH has showed that open dialogue and communication with all community stakeholder groups, including adolescents, has been effective [43,52]. Based on learning from HIV experiences, it is possible to address stigma in healthcare facilities by engaging in participatory dialogue and action planning with stakeholder groups [52,76]. A recent study in Burundi also recommended engaging religious leaders, confronting gender-based power imbalances and exploring the underlying drivers of health practitioner stigma as norm-shifting interventions [77]. Engaging adolescents and stakeholders in participatory action research focused on stigma reduction can help to address this pernicious barrier to improve ASRH outcomes [2,10,29,45,50,67,78].

The concepts identified in the literature on Kenya, Rwanda, Tanzania, and Uganda echo the concepts identified in the global SRH literature over the last 20 years, including family planning, contraceptives, STIs, puberty and abstinence [1,2,8,10,11,22,26,37,39,40,42–45,50,51]. Although important, these SRH concepts do not address a wider scope of ASRH perspectives that include sexual desires, masturbation, and sex for pleasure [10,49,79]. These topics have been included in "Comprehensive sexuality education" (CSE) programs implemented in some Sub-Saharan African countries after the 1994 Conference on Population and Development [80]. In East Africa, however, topics such as masturbation, abortion and sexual orientation directly contradict religious beliefs and traditions such as premarital sexual abstinence and patriarchal gender norms resulting in resistance from communities to include such topics in sex education [80,81]. Research has demonstrated that capturing a wider scope of SRH perspectives among adolescents is important to inform more effective, inclusive and adolescent context-specific research, programming and education [49].

The studies reviewed framed adolescent perspectives of SRH within assumptions of risky behaviour [8,11,43–45,50,52–55,58,61,66,67]. This narrow framing of adolescence as an inherently risky time in an individual's life does not recognize the intersectional experience of adolescents, ultimately limiting how societies support them [29,66,67]. Adolescence is a key time in which intersecting experiences of race, class, gender, and age impact the health of young people. An intersectional lens to ASRH recognizes the social, political and economic power structures that shape individual interactions and health experiences [82]. For example, economic vulnerability and unequal gender norms in East Africa result in transactional sex and limits agency among young women to access SRH services [8,10,38,42–44,50,53–55,82]. Recognizing the intersecting experiences specific to adolescence, rather than framing adolescence as inherently risky, invites an intersectional reframing of adolescence as a time of capacity building and growth towards increased engagement in SRH [2,10,29,45,50,67,78], leading to more targeted, effective ASRH research, programming and education [82].

This study had several limitations. First, literature was only included if written in English, limiting the diversity of perspectives captured. However, the majority of the literature involved authors from institutions based in East Africa in collaboration with institutions from the United States or Europe. Second, literature was only included if available through university accessible databases. However, the databases provided a wide scope of journals and articles. Finally, the data extraction and analysis of literature was completed only by the primary author (HC). Although we recognize that best practice involves multiple reviewers, the input from contributing authors on conceptualization (LB, HC, PO, DD), data collection (PO, HC), and drafting and editing (HC, DD, AB) added to the rigour of ideas shared.

## Conclusion

This scoping review has explored how adolescents have been engaged in SRH research in Kenya, Rwanda, Tanzania, and Uganda by identifying the methods used to gather adolescent perspectives of SRH, content of those perspectives and narratives framing discussions of ASRH. Findings suggest that there is opportunity for more meaningful engagement of adolescents in ASRH research and exploration of more diverse framings and concepts concerning ASRH, as identified by adolescents. Future research should explore how East African countries engage adolescents in the development of ASRH programming.

## Supporting information

**S1 Checklist. The 'Preferred Reporting Items for Systematic reviews and Meta-Analyses extension for Scoping Reviews (PRISMA-ScR) Checklist' was used in this study.** (DOCX)

**S1 Text. Search strategy.** A text file of the search strategy we used in this scoping review. (DOCX)

**S1 Data. A text file of the summarized data extracted.** (DOCX)

## Acknowledgments

We thank Dr. Lisa Schwartz for her support in the final stages of drafting the manuscript.

## Author Contributions

**Conceptualization:** Hanna Chidwick, Andrea Baumann, Laura Banfield, Deborah D. DiLiberto.

**Data curation:** Hanna Chidwick, Patricia Ogba.

**Formal analysis:** Hanna Chidwick, Deborah D. DiLiberto.

**Investigation:** Hanna Chidwick, Patricia Ogba.

**Methodology:** Hanna Chidwick, Laura Banfield, Deborah D. DiLiberto.

**Project administration:** Hanna Chidwick.

**Supervision:** Deborah D. DiLiberto.

**Validation:** Deborah D. DiLiberto.

**Writing – original draft:** Hanna Chidwick.

**Writing – review & editing:** Hanna Chidwick, Andrea Baumann, Deborah D. DiLiberto.

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
