## [Decision Letter · Decision Letter 0]

4 Jan 2022

PGPH-D-21-00836

Exploring adolescent engagement in sexual and reproductive health program and service development in East Africa: a scoping review

Dear Dr.Chidwick

Thank you for submitting your manuscript to PLOS Global Public Health. After careful consideration, we feel that it has merit but does not fully meet PLOS Global Public Health’s publication criteria as it currently stands. Therefore, we invite you to submit a revised version of the manuscript that addresses the points raised during the review process.

EDITOR:

The authors are expected to address the specific issues raised by the two reviewers and the handling editor.

While the  comments of reviewer 1 are appreciated, the submissions from the reviewer 2 and 3 and that of the handling editor posits that a minor review of the paper will improve the manuscript for acceptance and subsequent publication. This decision is arrived at on the basis that a theoretical review underpinning a scoping review is not necessarily justified to be grounds for an outright rejection of a work that contributes in the opinion of the two reviewers and the handling editor to the field of Adolescent Sexual and Reproductive Health. Rejection on the grounds of novelty is therefore considered not strong enough and not consistent with the

The authors must however address the theoretical issue raised by reviewer 1 as a matter of recommendation LOS Global Public Health’s publication criteria

We look forward to receiving your revised manuscript.

Kind regards,

Seth Christopher Yaw Appiah, PhD,PhD

Academic Editor

Journal Requirements:

1. Please provide separate figure files in .tif or .eps format only and remove any figures embedded in your manuscript file.  If you are using LaTeX, you do not need to remove embedded figures.

2. Please note that your Data Availability Statement is currently missing the repository name and/or the DOI/accession number of each dataset OR a direct link to access each database. If your manuscript is accepted for publication, you will be asked to provide these details on a very short timeline. We therefore suggest that you provide this information now, though we will not hold up the peer review process if you are unable.

Additional Editor Comments (if provided):

The authors have provided an important topic of comptemporary relevance to the world and particularity the East African region with relatively very youthful population. The question on why some East African Countries were excluded have not justifiably been explained. Authors will do a great benefit to readers by providing justification for or upgrading the scope to reflect the entire East African Countries. In going through appendix 2, it appear the authors in using the Boolen operators did not include countries such as Burundi, South Sudan, Ethiopia, Somalia and a few other East African Countries. This can be confirmed in lines 70-74 of the manuscript. While appendix 3 presents the summary of the findings, it will be better a section is created in the tale to indicate where each of the study is being conducted. A major limitation of this paper is the absence of 'where the study was conducted' as a central criteria informing the data extraction framework. The use of the lead authors affiliation may not suffice and may potentially lead to exclusion of several significant articles addressing the topic if the study setting/location is not included as a central criteria . These issues ought to be addressed significantly. The authors could equally re-align their title to reflect ' in selected countries in Africa' to reflect the accurate submission of the paper.

Reviewers' comments:

Reviewer's Responses to Questions

**Comments to the Author**

1. Does this manuscript meet PLOS Global Public Health’s publication criteria? Is the manuscript technically sound, and do the data support the conclusions? The manuscript must describe methodologically and ethically rigorous research with conclusions that are appropriately drawn based on the data presented.

Reviewer #1: No

Reviewer #2: Yes

Reviewer #3: Partly

2. Has the statistical analysis been performed appropriately and rigorously?

Reviewer #1: N/A

Reviewer #2: Yes

Reviewer #3: Yes

3. Have the authors made all data underlying the findings in their manuscript fully available (please refer to the Data Availability Statement at the start of the manuscript PDF file)?

Reviewer #1: No

Reviewer #2: Yes

Reviewer #3: Yes

4. Is the manuscript presented in an intelligible fashion and written in standard English?

Reviewer #1: Yes

Reviewer #2: Yes

Reviewer #3: Yes

5. Review Comments to the Author

Reviewer #1: The review is very descriptive. I miss a clear theoretical frame that guides the review process. In the results section I am left wondering the so what question. What can we do with this information. What policies and programs can benefit from this evidence? The scope is too broad hence the novelty of the evidence is thin.

Reviewer #2: This study aimed to describe and show the amount of engagement adolescents have in East Africa in topics related to ASRH. The authors did a great job in detailing how this population is involved in the creation and design of scientific research to obtain information about narratives, perspectives and information on ASRH in the region. As a reviewer from a third world country, this study resonated with me ad the results and conclusion are very similar to how ASRH is perceived. This study is a great example of how much more needs to be done to include the populations researchers are trying to study IN the research design and not only as participants, because after all, they are the experts. Also, I think it serves an example of how health policies and prevention/action campaigns should be created and addressed, using these populations as the main subjects.

Nonetheless, I think there are a few recommendations I would like to make to improve the reading and comprehension to this study:

1. The abbreviated term ASRH is in the abstract but is not mentioned in the introduction, I consider the authors should place the term here too before abbreviating it.

2. The objective of the research is clear. However, it would be helpful if it was specified in the title and the abstract that what the authors are trying to find through the review is how adolescents are engaged on ASRH research because at first sight, it just seems that they are engaged WITH ASRH, instead of the research specifically.

3. I consider it would be important if the authors could briefly explain why they chose East Africa as the research region. This could give the manuscript a little bit more context for the reader.

4. It caught my attention that “funding” was included in the Data extraction framework (Table 1), so I think if the authors will include it here, they could explain why this is part of the inclusion/exclusion criteria or why is it important for the review.

5. In page 5 of the manuscript, I do not consider it is necessary to include the reviewers’ initials, they could be eliminated unless the authors want the readers to know who contributed in what section of the review.

6. When mentioning which articles apply to which statement (e.g. content of perspectives section), I consider it would be more helpful to place them as footnotes so that the reader can later go to the references section to find them.

7. On page 8, the statement “The fear of infertility is connected to the high value placed on having children in East Africa” could use an explanation or small discussion on the reason the authors think or found out in this review why this is important.

8. I noted the references are not in alphabetical order, make sure they are before submitting the final version, thanks!

9. To make the paper more understandable, especially the methods section, establishing a little bit more clearly the inclusion/exclusion and how the studies were selected (although it is clear the main author and reviewers worked as a group) would be of great help.

Overall, this is a great manuscript, easy to read and understand!!! I was impressed with the results and as I said, it resonated with me so I can’t wait for it to be published and let people know what we can do as researchers to improve SRH conditions in children and adolescents around the world.

Reviewer #3: 1. Is there a compelling reason why only 4 East African countries were included in the scoping review?

2. The depiction of what constitutes ‘contents of perspectives for SRH’ is quite categorical. Any reference for this or is it an arbitrary determination?

3.Was there a consideration of educational level as a possible determinant of participation by adolescents by any of the studies reviewed?

4.Was substance abuse appraised as a factor by any of the studies reviewed?

5.Aside the stated role of the primary author in sole extraction and analysis of data, the role of the other authors is somewhat ambiguous. Some clarification needed.

6.Overall, a good work.

6. PLOS authors have the option to publish the peer review history of their article (what does this mean?). If published, this will include your full peer review and any attached files.

**Do you want your identity to be public for this peer review?** For information about this choice, including consent withdrawal, please see our Privacy Policy.

Reviewer #1: No

Reviewer #2: **Yes: **Marinés Mejía

Reviewer #3: No

---

## [Editor Report · Decision Letter 1]

23 Jan 2022

Exploring adolescent engagement in sexual and reproductive health research in Kenya, Rwanda, Tanzania, and Uganda: a scoping review

PGPH-D-21-00836R1

Dear  Hanna Chidwick

We're pleased to inform you that your manuscript has been judged scientifically suitable for publication and will be formally accepted for publication once it meets all outstanding technical requirements.

Within one week, you'll receive an e-mail detailing the required amendments. When these have been addressed, you'll receive a formal acceptance letter and your manuscript will be scheduled for publication.

An invoice for payment will follow shortly after the formal acceptance. To ensure an efficient process, please log into Editorial Manager at https://www.editorialmanager.com/pgph/ click the 'Update My Information' link at the top of the page, and double check that your user information is up-to-date. If you have any billing related questions, please contact our Author Billing department directly at authorbilling@plos.org.

Kind regards,

Seth Christopher Yaw Appiah, PhD PhD

Academic Editor

Additional Editor Comments

I find the comments addressed by the authors as very adequate and well responded to. Further, i consider the findings as promising in its advancement of Adolescent Sexual and Reproductive Health Service research in East Africa. After reading the revised manuscript one more time and interrogating the author responses /comments, i find the article suitable for publishing in this reputable journal. On the basis of this, i recommend for acceptance of your manuscript.